# Status and Factors Associated with Healthcare Choices among Older Adults and Children in an Urbanized County: A Cross-Sectional Study in Kunshan, China

**DOI:** 10.3390/ijerph17228697

**Published:** 2020-11-23

**Authors:** Yuxi Zhao, Linqi Mao, Jun Lu, Qi Zhang, Gang Chen, Mei Sun, Fengshui Chang, Xiaohong Li

**Affiliations:** 1Department of Health Policy and Management, School of Public Health, Fudan University, Shanghai 200032, China; 18211020043@fudan.edu.cn (Y.Z.); maolinqi1995@163.com (L.M.); lujun@shmu.edu.cn (J.L.); sunmei@fudan.edu.cn (M.S.); changfsh@fudan.edu.cn (F.C.); 2China Research Center on Disability Issues at Fudan University, Shanghai 200032, China; gchen@shmu.edu.cn; 3Key Laboratory of Health Technology Assessment, National Health Commission, Fudan University, Shanghai 200032, China; 4School of Community and Environmental Health, Old Dominion University, Norfolk, VA 23529, USA; qzhang@odu.edu; 5Department of Health Law and Inspection, School of Public Health, Fudan University, Shanghai 200032, China

**Keywords:** healthcare choices, urbanized county, older adults, children, China

## Abstract

As important unit for regional health planning, urbanized counties are facing challenges because of internal migrants and aging. This study took urbanized counties in China as cases and two key populations as objects to understand different populations’ intentions of choosing corresponding health service resources and to provide support for resource allocation. A cross-sectional study was conducted in Kunshan, a highly urbanized county in China, in 2016, among older adults aged 60 or over and children aged 0–6. Multinomial logistics models were used to identify the factors associated with healthcare choices. In this study, we found that income, distance of the tertiary provider, and migrant status were not associated with choices of tertiary healthcare outside county for children, while parents’ education level was. The responsiveness of the tertiary provider inside the county was lower than primary and secondary providers inside the county, while respondents were dissatisfied with the medical technology and medical facility for the tertiary inside the county compared to those of the tertiary provider outside the county. Significant differences existed in terms of the perception of different categories of institutions. To conclude, local governments should particularly seek to strengthen pediatric primary health services and improve the responsiveness of healthcare facilities to treat geriatric and pediatric diseases, which also bring significance to the developing countries in the process of urbanization.

## 1. Introduction

Like most developing countries, the primary challenge of medical delivery in the Chinese health system is the gap in resource allocation across areas, as most human and material resources are concentrated in developed regions [1] with high-quality medical resources, particularly metropolises [2]. In addition, owing to a lack of strict gatekeeper and referral system patterns within the three-tier healthcare system [3], people embrace the freedom to choose specialized facilities directly rather than using primary health facilities. According to the regulation of health reform launched by the Chinese government in 2009, each level health provider of the three-tier healthcare system is oriented toward different populations and regions. Community health center/stations and township health center/village clinics are at the primary level, while city secondary hospitals and county hospitals are at the secondary level, and overcrowded hospitals (mainly located in urban areas) are at the tertiary level [4]. This is particularly common in economically developed big cities with high-quality health services, such as Shanghai and Beijing [5,6]. At the same time, community health centers, which are expected to serve gatekeeping roles by providing first-contact care, are underutilized [7,8]. This situation does not accord with the positioning of the prescribed three-tier institution, as tertiary hospitals at the city, provincial, and national levels are equipped with specialized medical resources and highly specialized medical experts. Improving the patterns of patient flow is one of the priorities of health reform launched by the Chinese government in 2009. In this regard, the county is an important unit for regional health planning. In 2015, the central government focused on establishing mechanisms for first medical visits at grassroots-level healthcare facilities, with the goal of providing care, even for serious illnesses at the county level [9].

The two-child policy took effect in China on 1 January 2016, allowing couples to have two children [10]. With the implementation of this policy, China’s population is expected to reach 1.45 billion by 2030 [11], which brings a considerable demand for pediatric services. Nevertheless, due to the shortage and unavailability of pediatric resources, pediatricians are not adequately equipped to handle the baby boom. According to an estimation [12], the shortage of pediatricians in China has reached 200,000. Thus, understanding the preferences of patients is an important prerequisite for the rational allocation of medical resources. At the same time, health systems in urbanized counties are facing great challenges caused by the floating population (otherwise known as internal migrants), especially children, in addition to the universal challenges caused by population aging [13]. To inform further service developments and the creation of a more sustainable, cost-effective health system, there is an emerging need to better understand healthcare choices and examine the key associated factors: Whether the current health service system meets the health needs of certain populations and to what degree. Furthermore, the examination in this study is in the context of county-level tertiary hospitals, which differs from existing research, which have primarily focused on medical institutions [1,4].

There are numerous factors, including patient and household factors, medical provider factors, and context factors, that can be associated with healthcare choices. From the perspective of provider and context factors, a study among about 40 patients in England showed that the perception of provider responsiveness, considering factors such as convenience, waiting time, and confidence, is a strong motivating factor when choosing primary care [14]. Perceived professionally relevant factors (e.g., whether the physician is certified) [15,16] and doctor’s quantity also affected patients’ choices [17]. In Australia, studies have demonstrated that geographical factors were associated with patients’ healthcare decision-making and the use of healthcare services, while the quality of healthcare (e.g., reputation) of different hospitals might be more important [18]. Participants’ sociodemographic characteristics exerted a considerable influence on their healthcare decisions. A study in Greece found that the utilization of health services was mostly determined by health status rather than socioeconomic factors like medical insurance [19]. A study conducted in 14 tertiary hospitals in Shanghai, China, showed that patients’ healthcare-seeking preferences were influenced mainly by illness severity and sociodemographic characteristics, and patients who earned higher monthly incomes expressed a preference for first-class providers [20]. Based on the hypothesis that there are certain differences in the choice of healthcare providers between adults aged 60 or over and children aged 0–6, this study took developed counties in China as cases and two key populations as objects to understand different populations’ intentions of choosing corresponding health service resources and to provide support for resource allocation.

In China’s mainland, counties are defined as rural areas and the fundamental units responsible for both developing the economy and providing health services [21]. Counties are the basic unit for regional health planning, and with the improvement of the capacity of many hospitals in highly urbanized counties, they have been further upgraded to level-three hospitals [22] responsible for undertaking the diagnosis and treatment of common and frequently occurring diseases [23]. Thus, we selected Kunshan country, the most highly urbanized county in mainland China, to study people’s choices of healthcare provider types and the associated factors. As older adults and young children are the most vulnerable [24], we focused on healthcare-seeking patterns in these populations. Therefore, the aims of this study were to: (1) Describe and compare the choices of healthcare providers between older adults and young children; (2) Analyze the factors associated with healthcare decisions from three aspects: Choice of the highest (tertiary) healthcare providers outside county, choice of tertiary healthcare providers within county, and the perception—mainly based on responsiveness and the evaluation of medical-related items—among older adults and children.

## 2. Materials and Methods

### 2.1. Study Design and Setting

This was a cross-sectional study conducted in Kunshan County from June to August 2016. Kunshan County, closed to Shanghai and located in Suzhou City in the southeastern part of the Jiangsu Province, has topped the list of “Best Counties in China’s Mainland” by *Forbes China* from 2008 to 2014. Kunshan County has a total area of 928 square kilometers. The poverty line was yuan 7500 in 2016 [25]. Based on government statistics, the annual per capita gross domestic product (GDP) in Jiangsu was yuan 107,189 in 2017, placing the area fifth among 31 provinces in China (National Bureau of Statistics, 2017). The per capita GDP of Kunshan in 2017 was about yuan 212,100, while that of the whole mainland was yuan 59,660. By the end of 2015, the population in Kunshan was 1.69 million: 0.79 million local residents with Kunshan hukou and 0.90 million internal migrants without Kunshan hukou (also called floating people, having lived in Kunshan for more than 6 months). Of these, there were nearly 111,000 children aged 0–6 and 160,000 residents aged 60 or over. In this study, children aged 0–6 and older adults aged 60 or over formed the target population, and migrant residents refer to those who had migrated to Kunshan County and lived in Kunshan for at least 6 months (or from birth).

There are 2.55 medical practitioners per 1000 population in Kunshan [26], lower than 3.80 per 1000 population in Shanghai [27]. The number of pediatric medical practitioners was 112,800 in 2017 in China, accounting for only 3.9% of all medical practitioners [12]. There were 1.45 pediatricians per 1000 children <14 years in Shanghai [27], 1.55 per 1000 in Kunshan [26], and 0.49 per 1000 in underdeveloped counties in China [28]. There were 0.53 pediatricians per 1000 children <14 years at the national level [29], lower than the standard of 1.46 pediatricians in America [30]. In terms of the education level of health workers, about 43% doctors had college-level education or greater in Kunshan [26], the proportion in Shanghai was 37% [31].

This study was conducted after obtaining ethical approval (IRB#2015–TYSQ–03-11) from the Medical Research Ethics Committee, School of Public Health, Fudan University (IRB00002408 & FWA00002399). Completion of the survey was considered indicative of providing informed consent to participate.

### 2.2. Sampling

The participants were selected through a multistage stratified sampling framework. In the first stage, all 11 towns in Kunshan were divided into three tiers: Center, middle, and outer surroundings of the county. Then, three towns (KF, ZP, and JX) from the three tiers were identified as the research sites. During the second stage, we used the simple random sampling formula, mentioned below, to calculate the sample size, where δ = 0.05, u = 1.96, and α = 0.05:(1)n=ua22π(⊢π)δ2

Supposing that the proportion of choosing tertiary healthcare providers is 50% [32], we obtained a sample size of about 400 older adults and children, respectively. Finally, at the health service centers in the three selected communities, based on physical examinations for older adults, 650 older adults were queried about their choices in seeking healthcare and 625 completed the questionnaires, yielding a response rate of 96.15%. Based on the planned immunizations for children, 450 parents were surveyed and 428 completed the questionnaires, yielding a response rate of 95.20%.

### 2.3. Data Collection and Instruments

The survey was organized by the local Health Bureau in Kunshan in 2016. The investigators included teachers and graduate and undergraduate students from Fudan University, all of whom received formal training before the investigation.

The investigation items in this study were extracted from the “An Analysis Report of National Health Services Survey in China questionnaire” [33], the validity and reliability of which have previously been demonstrated [34]. In this study, six experts were invited to conduct the validity assessment. Prior to the formal survey, 51 participants (not included in the main analysis) were recruited for a preliminary investigation.

We conducted face-to-face interviews using a structured questionnaire based on the family unit (when the respondents were children, the children’s parents answered the questions). In terms of each family member, the main contents included basic individual characteristics such as age, gender, occupation, migrant status, education, medical insurance, and hospitalization. In terms of family information, the participants were asked about: (1) Basic family-level information, such as household size and annual income; (2) Their choices of different health facilities when older adults or young children were faced with three different degrees of illness: minor illness, an illness of unclear severity (somewhat serious), and serious illness. Minor illness included a common, frequently occurring ailment, such as a cold, cough, or chronic illness. Referring to “An Analysis Report of National Health Services Survey in China questionnaire,” serious sickness involved serious and unpleasant symptoms (e.g., sustained high fevers with breathing difficulties, chest pain, severe headache), while somewhat serious illness referred to chronic disease (e.g., diabetes). We first asked about choices of healthcare inside or outside the county. Then, we further inquired about the level of the facility (primary, secondary, or tertiary); (3) The participants who had medical services utilization or accompanying experience of outpatient (in 6 months) or inpatient services (in 1 year) responded to the responsiveness module of the World Health Survey (WHS) questionnaire and their evaluation of medical-related items (Cronbach’s α = 0.84).

### 2.4. Statistical Analysis

Double data entry was independently completed by two trained college students using EpiData 3.1 (EpiData Association, Denmark, Odense). Data analysis was performed with STATA 14.0 (Stata Corporation, College Station, TX, USA). Descriptive statistics and the χ^2 test were used to describe the sociodemographic characteristics of the older adults and children (Table 1) and differences between the healthcare choices of two groups (Table 2). We used a multinomial logistic model to identify the factors associated with choices of healthcare providers among older adults and children (Table 3 and Table 4). The dependent variables were different type of chosen hospitals inside and outside county, coded as: (i) Primary and secondary facilities inside the county, (ii) tertiary facilities inside the county, and (iii) tertiary facilities outside the county. The independent variables included annual per capita income, household size, nearest healthcare providers in Kunshan, migrant status, age, education, town, chronic diseases, and parents’ age and education. One-way ANOVA, for univariate analysis, was used to compare perception of different providers (Table 5). The overall test of relationship showed that the probability of the model was 0.0003 (<0.05, therefore significant). Odds ratios (OR) were calculated with 95% confidence intervals (CI). Statistical significance was defined as *p*-value < 0.05.

## 3. Results

### 3.1. Description of the Sample

Table 1 displays the sociodemographic characteristics of the surveyed older adults and children. Of all the participants, there were 625 older adults and 428 children, and nearly half of the children were migrants. Approximately 43% of older adults had primary school education, and 37% of children’s families had university-level education. Nearly half of the population had a medium level annual income. For most participants, the nearest provider was a primary facility.

### 3.2. Comparison of the Two Groups’ Healthcare Choices

As shown in Table 2, when the disease was serious, 30.84% of children chose tertiary healthcare facilities outside the county, which was much higher than older adults (19.20%). There was significant difference between choices of different healthcare providers at different conditions.

### 3.3. Multinomial Logit Estimates on Choices of Healthcare Providers among Older Adults

As depicted in Table 3, the significant factors were education level (OR = 0.600, *p* = 0.008), the type of nearest facility (OR = 0.149, *p* = 0.002), and the distance from the tertiary facility (OR = 3.874, *p* = 0.000) in the situation of minor or somewhat illness. As for tertiary facilities outside the county, when the nearest provider was tertiary (OR = 2.908, *p* = 0.005), older adults having higher education level (OR=1.450, *p* = 0.004) were more likely to choose the tertiary healthcare when facing serious illness.

### 3.4. Multinomial Logit Estimates on Choices of Healthcare Providers among Children

As depicted in Table 4, when experiencing a minor illness, families near the tertiary provider were less likely to prefer primary facilities inside the county (OR = 0.274, *p* = 0.012). Families with higher education level were more likely to choose tertiary providers outside the county when dealing with serious illness (OR = 1.500, *p* = 0.002). However, annual per capita income, the distance of the tertiary provider, and migrant status were not influencing factors in the choice of tertiary healthcare outside the county for children.

### 3.5. Comparing the Perception of Healthcare Services among Older Adults and Children

Table 5 shows the perception of health responsiveness and other medical items among older adults and children. The best-performing factors among health responsiveness inside the county were confidentiality and prompt attention (traveling) in both older adults and children. Both older adults and children indicated that the responsiveness of the tertiary provider was lower than primary and secondary providers inside the county, while both groups were dissatisfied with the medical technology and medical facility of the tertiary inside the county compared those of the tertiary outside the county.

## 4. Discussion

This is the first study in an urbanized county in China to examine the choices of healthcare providers inside and outside the county and the factors associated with those choices using a multinomial logistic model. In this study, nearly half of the young children were internal migrants, which proves that seeking healthcare is a great challenge among the floating population in China, especially for children. The survey data show that majority of older adults over 60 were non-migrants.

Older adults and young children are prioritized in China’s policy for basic public health services [6]. Compared with studies in other counties [35,36], the residents of Kunshan appeared to favor internal facilities to a greater degree. Kunshan has experienced high-speed urbanization, in the wake of which there have been sufficient resources for facility improvement. Primary facilities received more investment, while two secondary facilities were upgraded to tertiary hospitals. Owing to the strength of these local facilities, increasing numbers of patients have chosen to stay in the county for their healthcare needs. However, when we focused on the target population, differences in choices of healthcare facilities became clear. Compared with older adults (19.2%), 30.84% parents were more likely to prefer the highest-level healthcare resources outside the county when facing serious disease severity. This result is consistent with Zhou’s study [37]. Parents’ concern about their children’s health makes them prioritize high-level hospitals outside the county when economic conditions permit. Another important reason may be the lack of well-trained pediatricians in local healthcare facilities, even in tertiary hospitals inside the county. We may interpret these data to indicate that the perceived inability of health providers to address severe diseases pushes residents to seek healthcare from facilities outside the county. Older adults, on the contrary, may prefer primary healthcare facilities because of chronic diseases.

Our results showed that disease severity was associated with the choice of healthcare facility, regardless of whether the population in question consisted of older adults or children. The differences in the factors associated with this choice in the two groups were apparent. In general, we found that while healthcare choices were mainly associated with sociodemographic characteristics, the type of health provider available was also an important factor. For older adults, the nearest provider type was significantly associated with healthcare choices, while for children, parental education level was an important factor. With regard to choosing local facilities, the correlations were more complex with various degrees of illness. The proximity of healthcare providers was important for older adults.

Nearest provider type, education level, and illness severity were significantly associated with healthcare choices in older adults. It is interesting to note that when older adults were seriously ill, they were more likely to choose a facility outside Kunshan if the nearest provider within the county was a tertiary facility. We may interpret this as indicating that households near higher-level facilities frequently sought healthcare there for minor ailments, owing to their deep mistrust in these facilities’ ability to address more serious illnesses. This study also showed the family factors in older adults’ choices of health facilities. A longer distance to the nearest facility was associated with a lower likelihood of choosing higher-level facilities in the case of a minor or somewhat illness. The possible reason was that, when only slightly sick, older adults could visit health facilities independently. However, when seriously ill, they would choose higher facilities, considering the close relationship between patients and their relatives in China [38].

In this study, nearly half of the young children were internal migrants. The interplay of better economic conditions, lack of health insurance for children, and lack of trust in the quality of facilities in Kunshan led parents to seek healthcare outside the county. These findings are in line with those of a Nigerian study reporting that high-level facilities, such as government-owned general/teaching hospitals, were the most commonly chosen in any childhood illness episode [39]. We found that household annual income was not the main factor in some circumstances, which is inconsistent with other studies in China [8,40]. Further, these findings are odds with studies in Brazil and Belgium, where income was one of the most significant determinants of healthcare utilization [41,42]. A previous study of migrants indicated that this group had a lower socioeconomic status than local residents [2]. In this study, migrant status did not play an important role in the choice of tertiary healthcare provider inside the county. However, the local children were more likely to choose the tertiary provider outside the county. In this study, migrant status did not play an important role in the choice of the tertiary healthcare provider inside the county. However, the local children were more likely to choose the tertiary provider outside the county. The present results may be owing to Kunshan’s high economic level and diminishing differences between the local and migrant residents. As these are no longer important factors, the differences in choices may have been caused by the factors diminishing during the process of urbanization. The residents of Kunshan were less sensitive to health expenditures than those in less urbanized places. Significant differences existed in terms of the perception of different categories of institutions. It is worth noting that the responsiveness of the tertiary provider was lower than primary and secondary providers inside the county. In addition, in the comparison of the level-three institutions within the city and outside the city, two groups indicated that they were dissatisfied with the medical technology and medical facility inside the county compared with those of the tertiary outside the county. As well as giving a short-term lift to responsiveness [43], local governments should concentrate on the county provider’s performance for the local government, especially tertiary facilities in the county. In the process of urbanization in other countries, especially in developing countries [44], the phenomena has appeared among a growing floating population. This brings challenges to the urban health service system, and the government may ignore the series of problems brought by this.

This study has some limitations. First, owing to the large floating population in Kunshan County and the lack of detailed information about said population, it is difficult to use random sampling to perform tests based on the total population. Thus, the study sample selected at the physical examination and planned immunization services sites were, to some extent, representative of older adults and children living in Kunshan, respectively, which may have influenced the reflection of the real situation. Second, this study mainly concentrated on residents’ responses regarding their healthcare choices, which had some bias given their actual choices for treatment. Finally, as the study was only conducted in Kunshan, the generalizability of the results is limited. We will address these limitations in future studies.

## 5. Conclusions

Compared with older adults, parents are more likely to prefer highest-level local facilities outside the county for their offspring, especially for younger children. It appears that parents lack confidence in primary healthcare, even when the child’s illness is not serious. In addition, a substantial proportion of both older adults and parents preferred tertiary healthcare providers outside the county, owing to a lack of trust in their county counterparts’ ability to manage serious diseases. A substantial proportion of residents chose to seek healthcare outside the county, which means that local healthcare providers were faced with potential health service challenges, especially pediatric health services. Based on these results, local governments should particularly seek to strengthen pediatric primary health services and improve the responsiveness of healthcare facilities to treat geriatric and pediatric diseases. Meanwhile, tertiary healthcare providers inside the county should be improved. It is especially important to improve the responsiveness and ability of tertiary healthcare providers to deal with complicated pediatric diseases.

## Figures and Tables

**Table 1 ijerph-17-08697-t001:** Sociodemographic characteristics of older adults and children (%).

Variables	Older Adults (n = 625)	Children (n = 428)			
		Total	Local (n = 227)	Migrant (n = 201)	*p*-Value *
**Distance to tertiary provider**					0.002
Far (<10 km)	211 (33.8)	135 (31.5)	56 (24.7)	79 (39.3)	
Middle (10 km)	231 (37.0)	156 (36.4)	85 (37.4)	71 (35.3)	
Far (>20 km)	183 (29.2)	137 (32.0)	86 (37.9)	51 (25.4)	
**Education level (schooling years)**					0.007
Illiteracy (0 year)	245 (39.2)	7 (1.6)	6 (2.6)	1 (0.5)	
Primary school (6 years)	268 (42.9)	35 (8.2)	26 (11.5)	9 (4.5)	
Junior school (9 years)	76 (12.2)	127 (29.7)	62 (27.3)	65 (32.3)	
Senior high school (12 years)	34 (5.4)	102 (23.8)	45 (19.8)	57 (28.4)	
College, university and above (15 years)	2 (0.3)	157 (36.7)	88 (38.8)	69 (34.3)	
**Household size**					<0.001
1	60 (9.6)	-	-	-	
2	278 (44.5)	6 (1.4)	2 (0.9)	4 (2.0)	
3–4	113 (18.1)	214 (50.0)	89 (39.2)	125 (62.2)	
≥5	174 (27.8)	208 (48.6)	136 (59.9)	72 (35.8)	
**Annual per capita income**					0.514
Low (<yuan 15,000 yuan)	214 (34.3)	74 (17.3)	37 (16.3)	37 (18.4)	
Medium (yuan 15,000–29,999)	297 (47.5)	200 (46.7)	112 (49.3)	88 (43.8)	
High (≥yuan 30,000)	114 (18.2)	154 (36.0)	78 (34.4)	76 (37.8)	
**Nearest providers ****					0.613
Primary	513 (82.1)	314 (73.4)	171 (75.3)	143 (71.1)	
Secondary	77 (12.3)	93 (21.7)	46 (20.3)	47 (23.4)	
Tertiary	35 (5.6)	21 (4.9)	10 (4.4)	11 (5.5)	

* Differences in variables between local children and migrant children. ** Nearest providers: The type of nearest provider for respondents in Kunshan.

**Table 2 ijerph-17-08697-t002:** Choices of healthcare providers among older adults and children.

Variables	Condition						*p*-Value **
	Minor		Somewhat		Serious		
	N	%	N	%	N	%	
**Older adults (n = 625)**							
Primary facilities inside the county	463	74.08	175	28.00	56	8.96	<0.001
Secondary facilities inside the county	96	15.36	114	18.24	61	9.76	
Tertiary facilities inside the county	58	9.28	318	50.88	386	61.76	
Secondary outside the county	3	0.48	1	0.16	2	0.32	
Tertiary facilities outside the county	5	0.80	17	2.72	120	19.20	
**Children (n = 428)**							
Primary facilities inside the county	265	61.92	96	22.43	25	5.84	<0.001
Secondary facilities inside the county	97	22.66	53	12.38	22	5.14	
Tertiary facilities inside the county	57	13.32	242	56.54	247	57.71	
Secondary outside the county	0	0.00	1	0.24	2	0.47	
Tertiary facilities outside the county	9	2.10	36	8.41	132	30.84	
***p*-value ***	<0.001		<0.001		<0.001		

* Differences in choices of healthcare providers between two groups at different condition. ** According to each group, differences in choices of healthcare providers at different condition.

**Table 3 ijerph-17-08697-t003:** Multinomial logit estimates on choices of healthcare providers among older adults.

Characteristics	Older Adults (n = 625)					
	Minor Illness		Somewhat Illness		Serious Illness	
	OR (95%CI)	*p*-Value	OR (95%CI)	*p*-Value	OR (95%CI)	*p*-Value
	**Category 1 vs. Category 2 (base) ***					
Age	1.010 (0.962,1.060)	0.690	1.007 (0.982,1.032)	0.594	1.007 (0.977,1.038)	0.646
Gender	0.663 (0.325,1.353)	0.259	1.009 (0.693,1.470)	0.962	0.958 (0.607,1.512)	0.854
Education	0.600 (0.412,0.875)	0.008	0.645 (0.512,0.814)	<0.001	0.859 (0.645,1.145)	0.301
Chronic diseases	0.947 (0.473,1.894)	0.877	0.998 (0.699,1.426)	0.993	0.887 (0.575,1.368)	0.588
Annual per capita income	0.912 (0.578,1.437)	0.690	0.912 (0.710,1.173)	0.473	1.047 (0.770,1.424)	0.771
Household size (Ref: ≤4)						
≥5	0.600 (0.297,1.214)	0.155	0.707 (0.475,1.053)	0.088	0.697 (0.417,1.165)	0.168
Nearest providers (Ref: not tertiary)						
Tertiary	0.026 (0.011,0.065)	<0.001	0.149 (0.044,0.508)	0.002	0.185 (0.024,1.421)	0.105
Distance to tertiary (Ref: <10 km)						
Middle (10-km)	0.848 (0.417,1.722)	0.648	2.279 (1.500,3.462)	<0.001	1.039 (0.624,1.732)	0.883
Far (>20 km)	3.838 (1.042,14.133)	0.043	3.874 (2.430,6.178)	<0.001	0.872 (0.498,1.527)	0.631
	**Category 3 vs. Category 2 (base) ***					
Age	1.057 (0.924, 1.210)	0.417	1.014 (0.941,1.094)	0.708	1.004 (0.972,1.036)	0.815
Gender	0.210 (0.018, 2.468)	0.214	0.822 (0.267, 2.532)	0.733	1.071 (0.675,1.701)	0.770
Education	0.908 (0.282, 2.930)	0.872	1.983 (1.060,3.710)	0.032	1.450 (1.23,1.871)	0.004
Chronic diseases	2.500 (0.245,25.544)	0.440	1.292 (0.696,2.395)	0.417	0.846 (0.544,1.315)	0.456
Annual per capita income	0.865 (0.238,3.139)	0.825	0.953 (0.468,1.940)	0.895	1.392 (1.028,1.885)	0.032
Household size (Ref: ≤4)						
≥5	0.504 (0.048,5.266)	0.567	0.968 (0.318,2.947)	0.954	1.170 (0.731,1.874)	0.512
Nearest providers (Ref: Not tertiary)						
Tertiary	0.423 (0.036,5.012)	0.495	1.963 (0.394,9.780)	0.411	2.908 (1.387,6.095)	0.005
Distance to tertiary (Ref: <10 km)						
Middle (10-km)	1.196 (0.130 10.971)	0.874	1.694 (0.486,5.908)	0.408	0.842 (0.513,1.384)	0.498
Far (>20 km)	3.105 (0.163,59.325)	0.452	3.555 (0.917,13.785)	0.067	0.638 (0.352,1.156)	0.139

* category 1: Primary and secondary facilities inside the county; category 2: Tertiary facilities inside the county; category 3: Tertiary facilities outside the county.

**Table 4 ijerph-17-08697-t004:** Multinomial logit estimates on choices of healthcare providers among children.

Characteristics	Children (n = 428)					
	Minor Illness		Somewhat Illness		Serious Illness	
	OR (95%CI)	*p*-Value	OR (95%CI)	*p*-Value	OR (95%CI)	*p*-Value
	**Category 1 vs. Category 2 (base) ***					
**Age**	1.151 (0.930,1.426)	0.197	1.212 (1.040,1.412)	0.014	1.323 (1.051,1.664)	0.017
**Migrant (Ref: Not migrant)**						
Migrant	1.781 (0.945,3.357)	0.074	1.195 (0.752,1.898)	0.452	1.550 (0.758,3.169)	0.230
**Annual per capita income**	0.623 (0.397,0.979)	0.040	0.713 (0.522,0.974)	0.033	0.638 (0.398,1.024)	0.063
**Household size (Ref: ≤4)**						
≥5	0.744 (0.403,1.373)	0.344	1.050 (0.665,1.657)	0.835	1.390 (0.690,2.800)	0.357
**Nearest providers (Ref: Not tertiary)**						
Tertiary	0.274 (0.101,0.748)	0.012	0.421 (0.135,1.320)	0.138	0.476 (0.059,3.848)	0.486
**Parents’ age**	1.041 (1.000,1.083)	0.046	1.017 (0.991,1.044)	0.212	0.992 (0.955,1.031)	0.692
**Parents’ education**	1.115 (0.804,1.547)	0.514	1.028 (0.812,1.302)	0.818	0.800 (0.561,1.141)	0.218
	**Category 3 vs. Category 2 (base) ***					
**Age**	1.064 (0.597,1.900)	0.832	1.057 (0.816,1.369)	0.676	1.109 (0.947,1.300)	0.200
**Migrant (Ref: Not migrant)**						
Migrant	4.185 (0.827,21.168)	0.083	1.541 (0.722,3.290)	0.263	1.010 (0.631,1.619)	0.966
**Annual per capita income**	6.697 (0.823,54.470)	0.075	1.505 (0.858,2.642)	0.154	1.254 (0.903,1.743)	0.177
**Household size (Ref: ≤4)**						
≥5	0.265 (0.045,1.571)	0.144	1.426 (0.674,3.018)	0.353	1.582 (0.992,2.522)	0.054
**Nearest providers** **(Ref: Not tertiary)**						
Tertiary	1.178 (0.110,12.609)	0.892	0.862 (0.185,4.017)	0.850	1.492 (0.582,3.826)	0.405
**Parents’ age**	1.113 (1.007,1.231)	0.036	1.032 (0.986,1.081)	0.180	1.023 (0.994,1.052)	0.116
**Parents’ education**	1.614 (0.628,4.152)	0.320	1.415 (0.925,2.165)	0.109	1.500 (1.160,1.941)	0.002

* category 1: Primary and secondary facilities inside the county; category 2: Tertiary facilities inside the county; category 3: Tertiary facilities outside the county.

**Table 5 ijerph-17-08697-t005:** Comparing the perception of healthcare services among older adults and children.

Theme	cat.1	cat.2	cat.3	*p*-Value	*p*-Value		
					cat.1 vs. cat.2	cat.1 vs. cat.3	cat.2 vs. cat.3
	**Older adults (n = 625)**						
**Responsiveness**							
Communication	4.286	3.974	3.905	<0.001	<0.001	<0.001	0.387
Autonomy	4.286	3.965	3.905	<0.001	<0.001	<0.001	0.511
Confidentiality	4.494	4.299	4.270	<0.001	<0.001	0.001	0.954
Prompt attention (traveling)	4.682	4.465	4.143	<0.001	<0.001	<0.001	0.007
Prompt attention (waiting)	4.136	4.090	3.762	<0.001	0.559	0.001	0.006
Dignity	4.494	4.297	4.254	<0.001	<0.001	0.001	0.892
Choice	4.023	3.936	3.714	<0.001	0.005	0.050	0.229
Quality of basic amenities	4.042	4.032	3.819	<0.001	0.965	0.175	0.205
**Others**							
Medical technology	3.815	3.968	4.191	<0.001	<0.001	<0.001	<0.001
Medical facility	3.920	4.035	4.206	<0.001	0.002	<0.001	0.009
Medical charges	3.815	3.776	3.794	0.767	0.857	0.992	0.996
Medicine	3.686	3.968	4.095	<0.001	<0.001	<0.001	0.023
	**Children (n = 428)**						
**Responsiveness**							
Communication	4.371	3.982	3.899	<0.001	<0.001	<0.001	0.268
Autonomy	4.362	3.977	3.928	<0.001	<0.001	<0.001	0.635
Confidentiality	4.491	4.398	4.362	0.107	0.221	0.230	0.953
Prompt attention (traveling)	4.647	4.519	4.348	<0.001	0.019	0.002	0.130
Prompt attention (waiting)	4.069	4.153	3.812	<0.001	0.086	0.002	0.000
Dignity	4.491	4.407	4.377	0.175	0.323	0.307	0.967
Choice	3.966	3.982	4.058	0.361	0.974	0.601	0.730
Quality of basic amenities	3.966	3.982	4.058	0.361	0.974	0.601	0.730
**Others**							
Medical technology	3.565	4.000	4.362	<0.001	<0.001	<0.001	<0.001
Medical facility	3.707	4.000	4.275	<0.001	<0.001	<0.001	0.001
Medical charges	3.853	3.829	3.628	0.089	0.979	0.043	0.142
Medicine	3.582	4.120	4.188	<0.001	<0.001	<0.001	0.613

cat.1: Primary and secondary facilities inside the county; cat.2: Tertiary facilities inside the county; cat.3: Tertiary facilities outside the county.

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
