# Peer review of "Status and Factors Associated with Healthcare Choices among Older Adults and Children in an Urbanized County: A Cross-Sectional Study in Kunshan, China"

_ijerph, 2020, doi:10.3390/ijerph17228697_

Round 1

Reviewer 1 Report

I do not have any major reservations for this manuscript. However, the following are a couple of minor comments.

  1. Please mention the age limit for older adults (greater equal 60) in the abstract and introduction. 
  2. As this paper is very specific to one country's healthcare system and uses data of one county, I would recommend adding some discussion on how this study can be helpful to researchers from different countries. 

Author Response

I do not have any major reservations for this manuscript. However, the following are a couple of minor comments.

  1. Reviewer’s comment: Please mention the age limit for older adults (greater equal 60) in the abstract and introduction.

Authors' response: Abstract section, line 26, page1, and introduction section line 88-89, page2, we add the limit for older adults.

  1. Reviewer’s comment: As this paper is very specific to one country's healthcare system and uses data of one county, I would recommend adding some discussion on how this study can be helpful to researchers from different countries.

Authors' response: Introduction section, page 2 and discussion section, page10, we added the implication of this study to other rapidly urbanizing developing countries, help to understand different populations' intention of choosing corresponding health service resources and provide support for resource allocation.

Reviewer 2 Report

This is a very interesting article that I really liked to read. I think it can benefit of some minor improvements:

Introduction.

It is well written and clearly reflects the problem to study.

Materials and Methods.

—In Study and Setting, the cross-sectional study is well described. But there is a considerable selection bias as it was conducted in Kunshan county and the source of people is limited. This bias is explained by the authors, but I think the bias is more important than the authors describe, so they could describe it better in Limitations.

—In Sampling, the authors write that «Owing to the large floating population in Kunshan county and the lack of detailed information about said population, it is difficult to use random sampling to perform tests based on the total population», and « Samples selected at the physical examination and planned immunization services sites are, to some extent, representative of older adults and children living in Kunshan, respectively». I think these sentences could fit better in Limitations section.

—I think that the paragraph «With the health reform initiated in 2009, the Chinese Ministry of Health issued the “National Guide for Basic Public Health Services,” which included 12 categories in the package of basic public health services for residents, covering all children aged 0-6 and older adults aged 60 or over regardless of hukou status [25]. The information on residents’ basic health status was recorded by public health service organizations. According to the policy of basic public health services equalization, communities should provide physical examinations for people aged 60 and over and planned immunizations for children aged 0-6 living in Kunshan county, regardless of whether they are local or floating residents» would fit better in the Introduction.

—In line 145, the authors write that «Supposing the proportion of choosing tertiary healthcare providers being 50%». I think they could support this statement with a reference or a source.

—They also describe that «450 children’s parents were surveyed». This again introduces a possible bias, as parents do not always reflect the truth when answering about their children's health decisions. Some parents are afraid when taking decisions about their children and many times these parents take not very logical decisions that they can not clearly explain later, or they do not remember the reasons why they they took those decisions. This is an important aspect that perhaps could be considered.

Results.

They are well described.

Discussion.

—The authors write that «Compared with older adults, children’s parents were more likely to prefer the highest-level healthcare resources inside the city or outside the county, regardless of disease severity». This is important when interpreting results, as parents can have a different vision when taking decisions about their children's health.

—In Limitations, I think that the selection bias could be more detailed, as this is an important bias. It does not invalidate the study, but it must be clearly described.

—The authors write that «Based on these results, local governments should particularly seek to strengthen pediatric primary health services». I do not see a clear relation within these two aspects. It is possible that pediatric primary health services are strengthened, and parents continue doing exactly the same. I think the authors could explain better why they state this.

But overall, despite these minor aspects, I think this is a very good research.

Author Response

Materials and Methods.

  1. Editor’s comment: In Study and Setting, the cross-sectional study is well described. But there is a considerable selection bias as it was conducted in Kunshan county and the source of people is limited. This bias is explained by the authors, but I think the bias is more important than the authors describe, so they could describe it better in Limitations.

Authors' response: Limitation part of discussion section, we described the limitations due to selection bias in more details.

  1. Editor’s comment: In Sampling, the authors write that «Owing to the large floating population in Kunshan county and the lack of detailed information about said population, it is difficult to use random sampling to perform tests based on the total population», and « Samples selected at the physical examination and planned immunization services sites are, to some extent, representative of older adults and children living in Kunshan, respectively». I think these sentences could fit better in Limitations section.

Authors' response: We have move these sentences to the Limitations section.

  1. Editor’s comment: I think that the paragraph «With the health reform initiated in 2009, the Chinese Ministry of Health issued the “National Guide for Basic Public Health Services,” which included 12 categories in the package of basic public health services for residents, covering all children aged 0-6 and older adults aged 60 or over regardless of hukou status [25]. The information on residents’ basic health status was recorded by public health service organizations. According to the policy of basic public health services equalization, communities should provide physical examinations for people aged 60 and over and planned immunizations for children aged 0-6 living in Kunshan county, regardless of whether they are local or floating residents» would fit better in the Introduction.

Authors' response: We have move these sentences to the Introduction section.

  1. Editor’s comment: In line 145, the authors write that «Supposing the proportion of choosing tertiary healthcare providers being 50%». I think they could support this statement with a reference or a source.

Authors' response: Materials and Methods section, line 140, page 3, we added the reference about the sentence.

  1. Editor’s comment: They also describe that «450 children’s parents were surveyed». This again introduces a possible bias, as parents do not always reflect the truth when answering about their children's health decisions. Some parents are afraid when taking decisions about their children and many times these parents take not very logical decisions that they can not clearly explain later, or they do not remember the reasons why they they took those decisions. This is an important aspect that perhaps could be considered.

Authors' response: Materials and Methods section, line 156, page 4, we added the method of face-to-face interview when the respondents were children. At the same time, it’s also a type of bias, we added the information in the Limitations section.

Discussion.

  1. Editor’s comment: The authors write that «Compared with older adults, children’s parents were more likely to prefer the highest-level healthcare resources inside the city or outside the county, regardless of disease severity». This is important when interpreting results, as parents can have a different vision when taking decisions about their children's health.

Authors' response: Discussion section, line 251, page 9, we added the explanation of results about this sentence.

  1. Editor’s comment: In Limitations, I think that the selection bias could be more detailed, as this is an important bias. It does not invalidate the study, but it must be clearly described.

Authors' response: Limitation part of discussion section, we described the limitations due to selection bias in more details.

  1. Editor’s comment: The authors write that «Based on these results, local governments should particularly seek to strengthen pediatric primary health services». I do not see a clear relation within these two aspects. It is possible that pediatric primary health services are strengthened, and parents continue doing exactly the same. I think the authors could explain better why they state this.

Authors' response: Materials and methods part, line 123, page 3, we added the background of pediatric resources in the investigation area. Based on this information and results of this study, we can conclude that the local pediatrician resources in Kunshan are insufficient, it’s necessary to strengthen pediatric primary health services.

Reviewer 3 Report

The subject used in the manuscript is very important for tertiary healthcare providers inside the county.

This study was conducted in only China. It is necessary to present what this study can contribute and suggest to the international health care system.

Why should you compare older adults and young children? It is necessary to add its significance in the introduction part.

Literature review on the relationship between the dependent variable and the independent variable considered in this study is insufficient.

What is the ethical consideration in this study?

How did you collect data for children? Did children's parents respond?

In the Table 1, sociodemographic characteristics, a children's educational level cannot be clarified at high school or college. Also, the sum of the percentages are not 100% for each element for the elderly and the child (i.e. the distance to the higher education provider and the annual capital income for the elderly, and the distance to the higher education provider and the education level of the child).

What is strength of this study?

Author Response

  1. Editor’s comment: This study was conducted in only China. It is necessary to present what this study can contribute and suggest to the international health care system.

Authors' response: Introduction section, page 2 and discussion section, page10, we added the implication of this study to other rapidly urbanizing developing countries, help to understand different populations' intention of choosing corresponding health service resources and provide support for resource allocation.

  1. Editor’s comment: Why should you compare older adults and young children? It is necessary to add its significance in the introduction part.

Authors' response: Introduction section, line 101, page 3, we added the significance of choosing the two type of population.

  1. Editor’s comment: Literature review on the relationship between the dependent variable and the independent variable considered in this study is insufficient.

Authors' response: Introduction section, line 75, page 2, we added the literature review of the influencing factors of medical care, including the perspective of facility and physician and participants’ sociodemographic characteristics.

  1. Editor’s comment: What is the ethical consideration in this study?

Authors' response: Page 11, we added the Ethics statement section.

  1. Editor’s comment: How did you collect data for children? Did children's parents respond?

Authors' Response: Materials and Methods section, line 156, page 4, we added the method of face-to-face interview when the respondents were children, the children's parents answered the questions of survey.

  1. Editor’s comment: In the Table 1, sociodemographic characteristics, a children's educational level cannot be clarified at high school or college. Also, the sum of the percentages are not 100% for each element for the elderly and the child (i.e. the distance to the higher education provider and the annual capital income for the elderly, and the distance to the higher education provider and the education level of the child).

Authors' response: We have correct these errors.

  1. Editor’s comment: What is strength of this study?

Authors' response: 1)This study investigated a highly urbanized county with a high proportion of internal immigrants, help other rapidly urbanizing developing countries to understand different populations' intention of choosing health service and provide support for resource allocation. 2)Samples selected based on basic public health services can to some extent represent local and floating households. 3)This study analyzed choices of healthcare both outside the county and within it based on a comparison of two household types. 4) It is of great significance to improve the health level of vulnerable groups (older adults and young children).

Reviewer 4 Report

This paper sets out to report factors associated with health care usage in two age groups; children and older adults. The authors make a number of conclusions some of which cannot be drawn from their present work. 

Overall the paper lacks the data to draw what may be some important issues. First, it is not at all clear what is meant by primary, secondary or tertiary facilities. The authors must provide a definition. Indeed, to anyone outside China it is likely that the lack of an overview of health care arrangements within the country (and the extent to which county variation exists) is a considerable problem. 

Second, the authors classify disease according to seriousness. It would appear to be based on patient assessment. There is no indication on what the actual disease profile is. Simply using high fever or cold as a classification is far too simplistic. 

Third, the authors refer to such measure as quality, effectiveness and best performing. Their literature on all of these topics is very weak. Indeed the value of cross country comparisons (comments on for example Belgium, Dutch or Nigerian care systems are in the majority irrelevant to the current work) must be questioned.

Fourth, the authors state that there is a lack of well-trained (as opposed to undertrained?) pediatricians. What is the basis for this? The basis for this needs to be made. At no point is there any detailed information provided about the actual as opposed to self perceived health of the population. What exactly is the care workforce for the population?

Finally, the term immigrant has wider connotations and I think its usage g=here is inappropriate. The authors should consider alternative wording. 

Overall the authors need to reconsider what they are trying to achieve with their work. At present the findings cannot be substantiated indeed given their data source the present objectives cannot be met. 

Author Response

  1. Editor’s comment: First, it is not at all clear what is meant by primary, secondary or tertiary facilities. The authors must provide a definition. Indeed, to anyone outside China it is likely that the lack of an overview of health care arrangements within the country (and the extent to which county variation exists) is a considerable problem.

Authors' response: Introduction section, line 44, page 1, we added definition of the three-tier healthcare system in China.

  1. Editor’s comment: Second, the authors classify disease according to seriousness. It would appear to be based on patient assessment. There is no indication on what the actual disease profile is. Simply using high fever or cold as a classification is far too simplistic.

Authors' response: Materials and Methods section, line 163, page 4, we added the detailed description of different level diseases.

  1. Editor’s comment: Third, the authors refer to such measure as quality, effectiveness and best performing. Their literature on all of these topics is very weak. Indeed the value of cross country comparisons (comments on for example Belgium, Dutch or Nigerian care systems are in the majority irrelevant to the current work) must be questioned.

Authors' response: Introduction section, line 75, page 2, we revised the literature review of the influencing factors of medical care, including the perspective of facility and physician and participants’ sociodemographic characteristics.

  1. Editor’s comment: Fourth, the authors state that there is a lack of well-trained (as opposed to undertrained?) pediatricians. What is the basis for this? The basis for this needs to be made. At no point is there any detailed information provided about the actual as opposed to self perceived health of the population. What exactly is the care workforce for the population?

Authors' response: Materials and methods part, line 125, page 3, we added the background of pediatric resources in the investigation area, the current situation of workforce in this county.

  1. Editor’s comment: Finally, the term immigrant has wider connotations and I think its usage g=here is inappropriate. The authors should consider alternative wording.

Authors' response: We replaced the term “immigrant” by “migrant”.

Reviewer 5 Report

Dear Author,

The present research addresses a highly topical issue of major policy concern. The subject and the concept of the article are extremely interesting and require substantive discussion in scientific journals.

Unfortunately, the paper suffers from some considerable deficits in the way in which both the conceptual and the methodological framework is presented and described. Throughout the paper, I am unclear on what the methods being used are describing. This is a significant issue and impacts on the overall paper. In more detail:

Abstract: There is a need to improve the abstract

Lines 3-8: need rewriting. The style is conversational, and it is preferred to focus on interpreting the results than quoting values.

Introduction: I found it difficult to follow the author’s logic as regards the merge between introduction and literature review. I would revise to better reflect the aims and work of the paper as well as the structure in the Introduction Section.

p.2, line 53 - p.2, line 92. I would strongly suggest that this part moves to a “Study Area” Section.

The literature review is not as developed as I would expect to see in an academic journal article. A further review of the literature is needed as there are significant deficiencies identified, regarding the determinants of medical delivery.

Materials and Methods:

Line 117: “the poverty line…”  reference needed.

Line 155: “An Analysis Report of National 155 Health Services Survey in China questionnaire” please put quotation marks.

Line 157: “In this study”, is written two times.

Lines: 162-163: Information on internal immigrants should be placed in “2.1. Study design and setting

Please provide more information on data and the model specification. In academic writing, there is a very specific process of building and presenting a statistical model.

 Results

The findings are interesting, but the aforementioned deficiencies in the way data have been described and presented significantly lessen their impacts. The descriptions of the factors assume that the reader has some conception of the factors considered in the model, which has not been presented.

---

According to my opinion, the paper is interesting to read, but the conceptual framework and scientific methods and assumptions are not clearly outlined. A review of the writing is needed as there are numerous issues throughout and the writing style is conversational.

Best Regards

Author Response

  1. Editor’s comment: There is a need to improve the abstract. Lines 3-8: need rewriting. The style is conversational, and it is preferred to focus on interpreting the results than quoting values.

Authors' response: Line 21, page 1, we have rewritten the Lines 3-8 and revised other sentences in the Abstract section, focusing on interpreting the results and significances of this study.

  1. Editor’s comment: I found it difficult to follow the author’s logic as regards the merge between introduction and literature review. I would revise to better reflect the aims and work of the paper as well as the structure in the Introduction Section.

p.2, line 53 - p.2, line 92. I would strongly suggest that this part moves to a “Study Area” Section.

The literature review is not as developed as I would expect to see in an academic journal article. A further review of the literature is needed as there are significant deficiencies identified, regarding the determinants of medical delivery.

Authors' response: Introduction section, line 75, page 2, we revised the literature review of the influencing factors of medical care, including the perspective of facility and physician and participants’ sociodemographic characteristics.

  1. Editor’s comment:

1) Line 117: “the poverty line…”  reference needed.

2) Line 155: “An Analysis Report of National 155 Health Services Survey in China questionnaire” please put quotation marks.

3) Line 157: “In this study”, is written two times.

4) Lines: 162-163: Information on internal immigrants should be placed in “2.1. Study design and setting”

5) Please provide more information on data and the model specification. In academic writing, there is a very specific process of building and presenting a statistical model.

Authors' response:

1) Line 114, page 3, we added the reference.

2) Line 151, page 4, we added quotation marks.

3) This is a error, we have corrected it.

4) These sentences have moved to the study design and setting section.

5) In the Statistical Analysis section, we provide more information on data and the model specification.

  1. Editor’s comment: The findings are interesting, but the aforementioned deficiencies in the way data have been described and presented significantly lessen their impacts. The descriptions of the factors assume that the reader has some conception of the factors considered in the model, which has not been presented.

Authors' response: We mainly revised the introduction, method and conclusion of the article to make the description of conclusion clearly. Revised the literature view and added the background of health workforce, strengthened the argument of the article.

Round 2

Reviewer 4 Report

This paper is a resubmission of the original and in the majority the authors have addressed the concerns raised. However, a number of issues remain.

First, on lines 125 to 132, the authors make reference to workforce ratios found in America. The number of health care personnel in any country is strongly correlated to wealth and has little to do with disease levels. I would suggest that the authors might highlight variation within China at the county level which may make their subsequent work of greater value as opposed to making international comparisons, the differences in which are after all minimal.

The paragraph involving lines 156 and 173 n3eds work. There is some duplication and the reference for the work needs clarifying. Is this the authors own classification or have other studies used it. For example a cold could give rise to a fever (I am not sure what a ‘high’ fever is opposed to other ‘fevers’.).

On the educational level some of the cells are rather small and I would suggest that the individuals are either excluded or the classes merged. Given the enormous changes seen in China over the lifetime of the adults it may be better to provide an alternative classification for example based on the number of years schooling although I have every sympathy with the authors is the difficulties this may create. A note about the reforms seen over the lives of this study group may however be beneficial to the readership.

Overall the authors have made substantial improvements to the original.

Author Response

1.Reviewer’s comment: First, on lines 125 to 132, the authors make reference to workforce ratios found in America. The number of health care personnel in any country is strongly correlated to wealth and has little to do with disease levels. I would suggest that the authors might highlight variation within China at the county level which may make their subsequent work of greater value as opposed to making international comparisons, the differences in which are after all minimal.

Authors' response: Materials and Methods section, we add the distribution of paediatricians in developed and underdeveloped county to highlight the variation of health workforce within China at the county level.

2.Reviewer’s comment: The paragraph involving lines 156 and 173 needs work. There is some duplication and the reference for the work needs clarifying. Is this the authors own classification or have other studies used it. For example a cold could give rise to a fever (I am not sure what a ‘high’ fever is opposed to other ‘fevers’.).

Authors' response: Data collection and instruments section, we deleted the repetition.

The classification of three different degrees of illness was based on “An Analysis Report of National Health Services Survey in China questionnaire” (we cited it in references [33], line 153). Furthermore, we explained the concept of “fever” in more details.

3.Reviewer’s comment: On the educational level some of the cells are rather small and I would suggest that the individuals are either excluded or the classes merged. Given the enormous changes seen in China over the lifetime of the adults it may be better to provide an alternative classification for example based on the number of years schooling although I have every sympathy with the authors is the difficulties this may create. A note about the reforms seen over the lives of this study group may however be beneficial to the readership.

Authors' response: Many thanks to the comments of reviewer.

In table1, some of the cells on the educational level was small for older adults, but children had a relatively high rate, it’s difficult to exclude the individuals, thus we keep the original individuals. Furthermore, we added “schooling years” in table 1.

Reviewer 5 Report

Dear author,

the response to the review comments was satisfactory, thus I suggest the publication of the manuscript.

Author Response

Authors' response:

Many thanks to the comments of reviewer. We checked the full text and corrected some spelling mistakes and highlight these revisions.